# Triglyceride-glucose index (TyG index) is associated with a higher risk of colorectal adenoma and multiple adenomas in asymptomatic subjects

Ziyue Zhu[1], Thomas Yuen Tung Lam[2,3]*, Raymond Shing Yan Tang[3], Sunny Hei Wong[4], Rashid Nok Shun Lui[3], Simon Siu Man Ng[3], Samuel Yeung Shan Wong[5], Joseph Jao Yiu Sung[4]

1 Stanley Ho Big Data Decision Analytics Research Centre, The Chinese University of Hong Kong, Ma Liu Shui, Hong Kong, 2 The Nethersole School of Nursing, The Chinese University of Hong Kong, Ma Liu Shui, Hong Kong, 3 Institute of Digestive Disease, The Chinese University of Hong Kong, Ma Liu Shui, Hong Kong, 4 Lee Kong Chian School of Medicine, Nanyang Technological University, Singapore, Singapore, 5 The Jockey Club School of Public Health and Primary Care, The Chinese University of Hong Kong, Ma Liu Shui, Hong Kong

* thomaslam@cuhk.edu.hk

**Data Availability Statement:** The data analyzed in this study contains potentially identifying or sensitive patient information and are not available

## Abstract

### Hypothesis

The objective of this study is to evaluate the predictive ability of the TyG index for the presence of adenoma and multiple adenomas in an asymptomatic population.

### Design

A secondary analysis was conducted on a prospective cohort of asymptomatic subjects aged between 50 and 75 who underwent CRC screening. Fasting blood glucose (FBG) and lipid profiles were measured within three months prior colonoscopy. TyG index was estimated as ln [fasting triglycerides (mg/dL) × FBG (mg/dL)/2]. Multivariate logistic regression was performed to assess the association between the TyG index and the risk of adenoma. Its association with multiple adenomas (≥5) and the continuous number of adenomas were assessed by multinomial regression and log-normal linear regression, respectively.

### Results

A total of 1,538 subjects were recruited among which 876 subjects (57%) had at least one adenoma detected. Elevated TyG index was positively associated with the incidence of adenoma (adjusted odds ratio [aOR]: 1.26, 95% confidence interval [CI]: 1.04–1.54). Compared with the lowest TyG index (≤ 8) group, the risk of adenoma was the highest among subjects in the highest TyG index (> 10) group (aOR: 3.36, 95% CI: 1.44–7.73). As compared to the non-adenoma group, the TyG index was also positively associated with multiple adenomas (aOR: 1.74, 95% CI: 1.17–2.57), and the estimate was also the highest in the highest TyG group (aOR: 14.49, 95% CI: 3.12–67.20). As for the number of adenomas, the positive

publicly because of the restrictions imposed by the Joint Chinese University of Hong Kong-New Territories East Cluster Clinical Research Ethics Committee. Upon request, de-identified data can be made available to researchers who meet the criteria for access to confidential data (email: crec@cuhk.edu.hk).

**Funding:** The studies are fully funded by the Hong Kong Jockey Club Charities Trust (Ref. no.: 2018/0043). URL: https://charities.hkjc.com/charities/english/charities-trust/index.aspx. The funders had no role in study design, data collection and analysis, decision to publish, or preparation of the manuscript.

**Competing interests:** The authors have declared that no competing interests exist.

**Abbreviations:** ADR, Adenoma detection rate; aOR, Adjusted odds ratios; AUC, Area under the curve; BMI, Body mass index; CDARS, Clinical Data Analysis and Reporting System; CI, Confidence intervals; CRC, Colorectal cancer; FBG, Fasting blood glucose; FIT, Fecal immunochemical test; FTG, Fasting triglycerides; HOMA-IR, Homeostatic Model Assessment of Insulin Re; IGF-1, Insulin-like growth factor-1; IGFBP, Inhibit IGF-binding protein; MetS, Metabolic Syndrome; ROC curve, Receiver operating characteristics curve; TyG index, Triglyceride-glucose index; WHO, World Health Organization.

association was maintained (Estimates: 1.06, 95% CI: 1.01–1.12) while the number of adenomas increase the most in the highest TyG index group (Estimates: 1.35, 95% CI: 1.10–1.65).

## Conclusions

Elevated TyG index is associated with an increased risk of colorectal adenoma and an increased number of adenomas for asymptomatic subjects aged $\geq$50.

## Trial registration

This study was registered on clinicaltrials.gov (NCT03597204 and NCT04034953).

## Background

Colorectal cancer (CRC) is the third most common cancer worldwide and the second most common cause of cancer death [1]. According to the World Health Organization (WHO)'s GLOBOCAN database, over 1.9 million new CRC cases were diagnosed and 930,000 CRC-related deaths were estimated in 2020 [2]. Fecal immunochemical test (FIT) and colonoscopy are recommended for CRC screening because of their effectiveness in reducing CRC incidence and mortality [3]. Among these methods for CRC screening, colonoscopy is proven to be more effective to identify more adenomas when compared to stool tests [4–6]. Most colorectal cancers start as a precancerous adenomatous polyp via the adenoma-carcinoma sequence [7]. Therefore, the removal of adenoma during colonoscopy can prevent the development of CRC [8]. Additionally, an increased adenoma detection rate (ADR) is associated with a reduced risk of CRC incidence and death [9, 10], and hence it is essential to identify subjects with a higher risk of adenoma and arrange them for colonoscopy.

Metabolic Syndrome (MetS) is a cluster of conditions that includes obesity, impaired glucose tolerance, hypertension, and dyslipidemia. It has been well documented that MetS is associated with a higher risk of CRC [11–14]. In particular, some studies have provided evidence that insulin resistance is associated with the presence of colorectal adenoma [15, 16]. The triglyceride–glucose (TyG) index is a solid biomarker of insulin resistance and is calculated with fasting blood glucose (FBG) and fasting triglycerides (FTG) [17], and it has been proven to be capable of predicting hypertension [18], cardiovascular diseases [19, 20], and CRC [21, 22]. However, there is no evidence that the TyG index is associated with the incidence of colorectal adenoma. In this study, we hypothesize that the TyG index may also predict the incidence of colorectal adenoma. Moreover, a higher TyG index may result in a larger number of adenomas for subjects. The association between the TyG index and colorectal adenoma and number of adenomas is evaluated among an asymptomatic population.

## Materials and methods

### Study design

The current study is a secondary data analysis of two prospective cohorts enrolled in the same charity-sponsored community-based CRC screening program, which was established in Hong Kong in 2008. The study settings of the two studies have been described elsewhere [23, 24]. In brief, subjects were recruited across the territories through conventional media (newspaper, radio, and television), social media promotion, and advertisement for CRC screening from

2008 to 2017 and for CRC and MetS screening from 2018 to 2023. Subjects of the 2008–2017 cohort were offered FIT, followed by a colonoscopy regardless of their FIT results; while subjects of the 2018–2023 cohort were offered FIT and MetS screening (fasting blood glucose and lipid profile tests) in the initial visit, and subjects with positive FIT results were referred to undergo colonoscopy within three months of the initial visit. Since no MetS screening is provided for the 2008–2017 cohort, retrospective data collection was conducted to collect their FBG and FTG blood results three months prior colonoscopy from a territory-wide electronic medical health record system, Clinical Data Analysis and Reporting System (CDARS), under the management of the Hospital Authority, the sole public healthcare provider in Hong Kong. CDARS is an anonymized electronic database that contains patients' demographic, procedural, medical and drug history, and laboratory results of all public hospitals and clinics in Hong Kong [25].

## Study population

The two studies have the same inclusion and exclusion criteria, and hence we do not differentiate the subjects from two cohorts. Asymptomatic and screening naïve subjects of age 50–75 were recruited. Exclusion criteria for the CRC screening programs included CRC symptoms such as per rectal bleeding, change in bowel habit, anemia, or weight loss, etc.; personal history of CRC, adenoma, inflammatory bowel disease, or colectomy; having received FIT in the past 2 years; flexible sigmoidoscopy in the past 5 years; colonoscopy in the past 10 years; strong family history of CRC (two or more first degree relative diagnosed CRC); medical conditions which were contraindications for colonoscopy. Subjects with available colonoscopy results were included in this secondary data analysis. Lastly, subjects with missing data, such as demographics, FBG and FTG were excluded from our analysis. The population in this study can largely represent the general population in Hong Kong because it shared a similar population distribution of the territory due to the widespread coverage of media promotion [24].

The two studies were approved by the Joint Chinese University of Hong Kong–New Territories East Cluster Clinical Research Ethics Committee (CRE-2015.468 approved on 7[th] October 2015, and CRE-2018.495 approved on 24[th] April 2018) and were registered on clinicaltrials.gov (NCT03597204 and NCT04034953). All of the participants signed the informed consent form.

## Data collection

All eligible subjects were invited to fill in a self-administered, structured questionnaire during the initial visit. The questionnaire recorded potential risk or protective factors for colorectal adenoma and cancer, including age and gender [26], overweight [27, 28], family history of CRC [26], cigarette smoking [29, 30], alcohol drinking [31], existing diabetes [32], and aspirin consumption [33–36]. In our study, body mass index (BMI) $\geq 23$ kg/m$^2$ was defined as overweight according to the Asian definition of the WHO [37]. TyG index was estimated as ln [FTG (mg/dL) × FBG (mg/dL)/2]. The normal range of the TyG index is considered below 8 [38]. In addition to the continuous TyG index, instead of using the sample quartiles and losing the generalizability, we categorized it as an ordinal variable: (i) < 8 (ii) 8–9 (iii) 9–10 (iv) > 10 [39]. Adenoma detection was defined as the detection of at least one adenoma, advanced adenoma, or colorectal cancer during colonoscopy. According to international guidelines, subjects with 5 or more adenomas detected during colonoscopy are considered to be at higher risk and recommended to have a surveillance colonoscopy in three years [40]. Therefore, $\geq 5$ adenomas were defined as multiple adenomas in our study.

## Outcome measures

The primary outcome of this study is the association of the TyG index and adenoma detection during CRC screening. The secondary outcome is the association of the TyG index and multiple adenomas ($\geq$5), and the continuous number of adenomas.

## Statistical analysis

Subjects' demographic factors were expressed as mean (standard deviation) or number (percentage) and were stratified by adenoma or no adenoma detection group. To evaluate the differences between the two groups, we used the student t-test and Pearson's chi-squared test to compare continuous and categorical variables, respectively.

For the estimations using detections as the endpoint, we performed univariate and multivariate logistic regression to assess the association of the TyG index and risk of colorectal adenoma and cancer without and with adjustment of different covariates (age, gender, overweight, cigarette smoking, alcohol consumption, diabetes, family history of CRC, and aspirin use), respectively. Age entered the estimations as a continuous variable, while the other covariates were dummy variables (for example, overweight = 1 if the subject was overweight, and 0 otherwise). For the estimations using multiple adenomas as the endpoint, we performed univariate and multivariate multinomial logistic regression without and with adjustment of covariates. The reference group for the multinomial regression was the non-adenoma detection group, and two comparison groups were the 1–4 adenomas detected group and the over 5 adenomas detected group. Odds ratios, adjusted odds ratios (aOR), and 95% confidence intervals (CI) were presented in the result tables. For the estimations using number of adenomas as the endpoint, since we had a heavy tail distribution for the number of adenomas, we transformed the dependent variable onto the log scale and applied log-normal linear regression. The estimates would be interpreted as a percentage change in the number of adenomas. Estimated coefficients and 95% CIs were presented for log-normal linear regressions without and with adjusted covariates.

Additionally, the covariate-adjusted receiver operating characteristics (ROC) curve calculated the area under the curve (AUC) was employed to test the ability of the TyG index for predicting the risk of having colorectal adenoma and multiple adenomas. The Hosmer–Lemeshow goodness-of-fit test was used to evaluate the reliability of the model, where $P > 0.05$ shows a good match of predicted risk over observed risk. For all analyses, a two-sided p-value $< 0.05$ was considered statistically significant. Statistical analyses were carried out with R version 4.2.2.

## Results

### Baseline characteristics

Fig 1 shows the flow chart of the patient inclusion and exclusion. A total of 1,538 subjects (mean age 60.5 [SD: 5.38]; 56.6% male gender) were included in the final analyses. Table 1 shows the subjects' baseline characteristics stratified by if adenoma was detected and removed during CRC screening. 876 (57%) subjects had at least one adenoma detected. Subjects with adenoma had higher mean TyG index, age, and BMI. They were also more likely to be male, smokers (current or past), drinkers, and have positive FIT test results. There was no difference in diabetes, family history of CRC, and the use of Aspirin between the two groups.

### Incidence of adenoma and CRC

Table 2 shows the unadjusted and adjusted odds ratios and the corresponding 95% CI for the association between continuous and categorical TyG index and colorectal adenoma risk. TyG

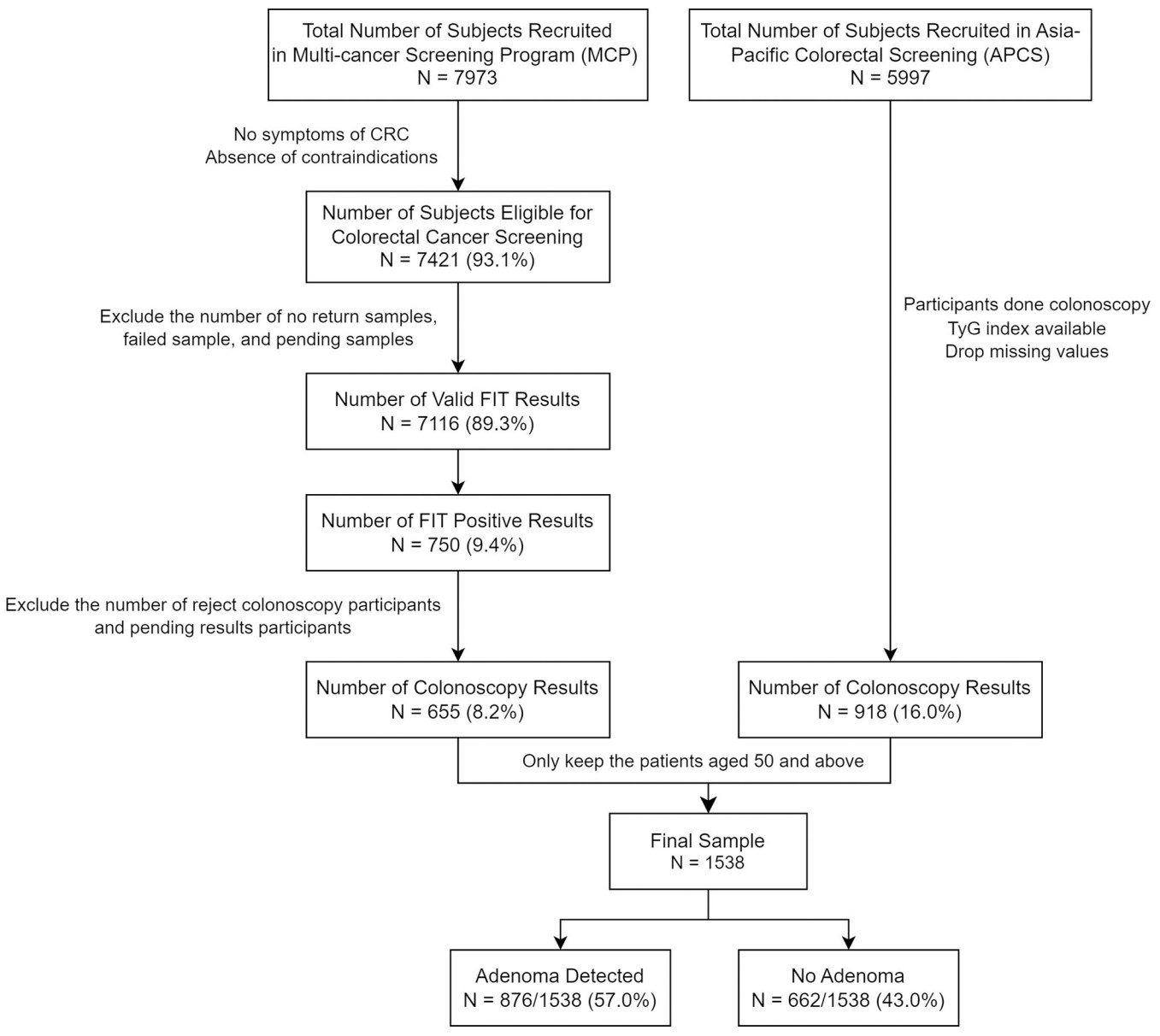

**Fig 1. Flow chart for the patient selection.**

index was statistically significantly associated with adenoma detection in both unadjusted and adjusted estimations. Every 1 unit increase in the TyG index would result in 1.26 times more likely for adenoma detection after adjusting confounders (CI: 1.04–1.54, p = 0.019). Compared to the group where a TyG index was less than 8, the subjects in the group with a TyG index ranged 9–10 were 1.52 times more likely to be detected with adenomas (CI: 1.02–2.25, p = 0.040); while the subjects in the group with TyG index ranged over 10 were 3.36 times more likely to be detected adenomas (CI: 1.44–7.73, p = 0.005). However, there were no statistically significant results for the group with a TyG index ranged 8–9 as compared to the reference group. The sensitivity, specificity, PPV and NPV of adenoma detection were presented in Table 3. Note that this model was not built for prediction and only aimed for showing the correlation between TyG index and risk of colorectal adenoma, and thus the values for sensitivity,

**Table 1. Baseline characteristics between subjects with and without colorectal adenoma.**

| | Adenoma | No Adenoma | |
|---|---|---|---|
| | n = 876 | n = 662 | p-value |
| TyG index, mean (SD) | 8.67 (0.63) | 8.60 (0.57) | 0.014 |
| Age, years mean (SD) | 61.0 (5.46) | 59.9 (5.21) | < 0.001 |
| Male sex, n (%) | 569 (65.0%) | 301 (45.5%) | < 0.001 |
| BMI, mean (SD) | 25.9 (3.82) | 24.8 (3.70) | < 0.001 |
| Overweight, n (%) (BMI $\geq$23 kg/m$^2$) | 687 (78.4%) | 449 (67.8%) | < 0.001 |
| Cigarette smoking (current or past), n (%) | 137 (15.6%) | 43 (6.5%) | < 0.001 |
| Alcohol consumption, n (%) | 150 (17.1%) | 52 (7.9%) | < 0.001 |
| Diabetes, n (%) | 183 (20.9%) | 129 (19.5%) | 0.516 |
| Family history of CRC, n (%) | 154 (17.6%) | 102 (15.4%) | 0.251 |
| Aspirin use, n (%) | 29 (3.3%) | 17 (2.6%) | 0.450 |
| FIT positive, n (%) | 678 (77.4%) | 277 (41.8%) | < 0.001 |

BMI: body mass index; CRC: colorectal cancer; FIT: fecal immunochemical test

specificity, PPV, and NPV were not sufficient to show enough discrimination power. Fig 2 shows the ROC curve for the adjusted categorical TyG index model and the AUC was 0.74 (CI: 0.72–0.77), presenting an acceptable discrimination power. The p-value of the Hosmer–Leme-show goodness-of-fit test was 0.425, which was larger than 0.05, and thus the model was reliable.

## Multiple adenomas

To analyze if the risk of different levels of the number of adenomas would respond differently to the TyG index, we categorized the number of adenomas into three groups: non-adenoma, 1–4 adenomas, and 5 or more adenomas [40]. Table 4 presents the estimation results. As compared to the non-adenoma group, a higher TyG index was statistically significantly associated with the risk being in the multiple adenomas group (5 or more adenomas) in both unadjusted and adjusted estimations. More specifically, every one unit increase in the TyG index would result in 1.74 times more likely to be in the multiple adenomas group after adjusting confounders (CI: 1.17–2.57, p = 0.006). Compared to the group where a TyG index was less than 8, the subjects in the group with a TyG index ranged 9–10 were 3 times more likely to be in the multiple adenomas group versus the non-adenoma group (CI: 1.04–8.61, p = 0.042); while for the subjects in the group with TyG index ranged over 10 were 14.49 times more likely to be in the

**Table 2. Association of TyG index and risk of colorectal adenoma.**

| | Unadjusted | | Adjusted* | |
|---|---|---|---|---|
| | OR (95% CI) | p-value | AOR (95% CI) | p-value |
| **TyG index (per 1 unit)** | 1.23 (1.04–1.46) | 0.015 | 1.26 (1.04–1.54) | 0.019 |
| **Ordinal TyG index** | | | | |
| $\leq$ 8 | Ref. | | Ref. | |
| 8–9 | 1.32 (0.98–1.79) | 0.071 | 1.36 (0.96–1.91) | 0.080 |
| 9–10 | 1.53 (1.08–2.17) | 0.016 | 1.52 (1.02–2.25) | 0.040 |
| > 10 | 2.73 (1.26–5.92) | 0.011 | 3.36 (1.44–7.73) | 0.005 |

*Adjusted for age, gender, overweight, smoking status, drinking status, diabetes, family history of CRC, aspirin use, and FIT test results.

**Table 3. Sensitivity, specificity, PPV and NPV for detecting adenoma and multiple adenomas (covariate-adjusted).**

|  | Sensitivity | Specificity | PPV | NPV |
|---|---|---|---|---|
| Adenoma Detection | 0.74 | 0.45 | 64.29 | 56.93 |
| Multiple Adenoma Detection | 0.01 | 0.99 | 20.00 | 94.52 |

PPV: Positive Predictive Value; NPV: Negative Predictive Value; The threshold of adenoma detection is 0.5, while the threshold of multiple adenomas is 0.33.

multiple adenomas group versus non-adenoma group (CI: 3.12–67.20, p < 0.001). As for the 1–4 adenomas detected group, there were undermined effects for both the magnitude and significant level of the estimates, showing a smaller disparity between the 1–4 adenoma group and the non-adenoma group as compared to the disparity between the multiple adenomas group and the non-adenoma group. Every one unit increase in the TyG index would result in 1.23 times more likely to be in the 1–4 adenomas group (CI: 1.01–1.50, p = 0.035); while for the subjects with TyG index ranged over 10 were 3.29 times more likely to be in the 1–4 adenomas group versus non-adenoma group (CI: 1.41–7.68, p = 0.006). The sensitivity, specificity, PPV and NPV of multiple adenomas detection were presented in Table 3. Given the low prevalence of multiple adenomas, the prediction was not precise. Even though the confusion matrix

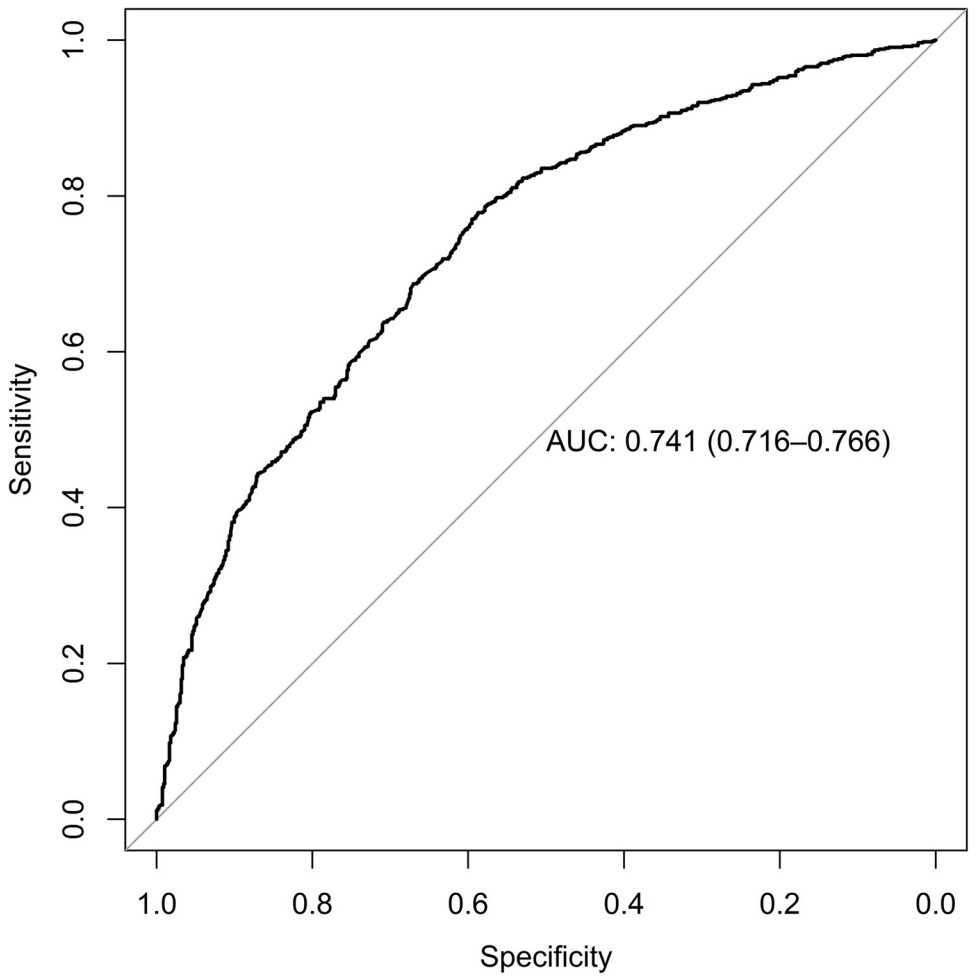

**Fig 2. Area under the curve for predicting adenoma incidence (covariate-adjusted).**

Table 4. Association of TyG index and multiple adenomas.

| | Unadjusted | | Adjusted* | |
|---|---|---|---|---|
| | OR (95% CI) | p-value | AOR (95% CI) | p-value |
| | 1–4 Adenomas | | | |
| TyG index (per 1 unit) | 1.16 (0.98–1.38) | 0.084 | 1.23 (1.01–1.50) | 0.035 |
| Ordinal TyG index | | | | |
| ≤ 8 | Ref. | | Ref. | |
| 8–9 | 1.22 (0.89–1.65) | 0.211 | 1.27 (0.91–1.78) | 0.164 |
| 9–10 | 1.39 (0.97–1.98) | 0.07 | 1.44 (0.98–2.14) | 0.066 |
| > 10 | 2.53 (1.14–5.62) | 0.022 | 3.29 (1.41–7.68) | 0.006 |
| | Over 5 Adenomas | | | |
| TyG index (per 1 unit) | 1.76 (1.25–2.48) | 0.001 | 1.74 (1.17–2.57) | 0.006 |
| Ordinal TyG index | | | | |
| ≤ 8 | Ref. | | Ref. | |
| 8–9 | 2.55 (0.99–6.53) | 0.052 | 2.61 (0.98–6.97) | 0.056 |
| 9–10 | 3.37 (1.24–9.15) | 0.017 | 3.00 (1.04–8.61) | 0.042 |
| > 10 | 10.70 (2.64–43.34) | < 0.001 | 14.49 (3.12–67.20) | < 0.001 |

*Adjusted for age, gender, overweight, smoking status, drinking status, diabetes, family history of CRC, aspirin use, and FIT test results

showed insufficient discrimination power, the Multi-class AUC was 0.70, presenting an acceptable discrimination power.

## Number of adenoma

In addition to exploring the categorization of multiple adenomas, we further examined if the TyG index was also associated with the continuous number of adenomas. The estimation results are presented in Table 5. The results remained significant, and the ordinal pattern persisted in continuous endpoint estimations. After adjusting confounders, every 1 unit increase in the TyG index would result in a 1.06% increase in the number of adenomas detected (CI: 1.01–1.12, p = 0.016). As compared to the reference group (TyG index less than 8), the subjects in the group that a TyG index ranged 9–10 had 1.11% more adenomas (CI: 1.01–1.23, p = 0.038); while for the group with the highest TyG index, the subjects were with 1.35% more number of adenomas (CI: 1.10–1.65, p = 0.004). In line with our former estimations, the estimates for the second group maintained insignificant.

Table 5. Association of TyG index and number of colorectal adenomas.

| | Unadjusted | | Adjusted* | |
|---|---|---|---|---|
| | Estimates (95% CI) | p-value | Estimates (95% CI) | p-value |
| TyG index (per 1 unit) | 1.07 (1.01–1.13) | 0.012 | 1.06 (1.01–1.12) | 0.016 |
| Ordinal TyG index | | | | |
| ≤ 8 | Ref. | | Ref. | |
| 8–9 | 1.09 (0.99–1.20) | 0.072 | 1.08 (0.99–1.18) | 0.085 |
| 9–10 | 1.15 (1.03–1.28) | 0.014 | 1.11 (1.01–1.23) | 0.038 |
| > 10 | 1.33 (1.07–1.66) | 0.010 | 1.35 (1.10–1.65) | 0.004 |

*Adjusted for age, gender, overweight, smoking status, drinking status, diabetes, family history of CRC, aspirin use, and FIT test results.

## Subgroup analysis

Since age and sex were the only two covariates whose estimates were both statistically significant in logistic regression, multinominal logistic regression, and log-normal linear regression, we conducted subgroup analysis by sex (male versus female) and age (aged < 60 versus ≥ 60). Evidence has proven that compared to the younger groups, the 60 years and older age group has a higher risk of colorectal adenoma [41, 42], and thus we chose 60 as the threshold to categorize ages. The models we adopted was the covariates adjusted models using adenoma detection, multiple adenomas detection, and number of adenoma as the endpoints, separately. The results were illustrated in Fig 3. In general, the higher the TyG index, the larger its effects on the risk of adenoma for each subgroup. Specifically, subjects who were female with a high TyG index (over 10) had a significantly higher risk of 1–4 colorectal adenomas as compared to males, while male with a high TyG index (over 10) had a significantly higher risk of multiple adenomas. There is no significant difference between the younger group (aged < 60) and older group (≥ 60) in our sample.

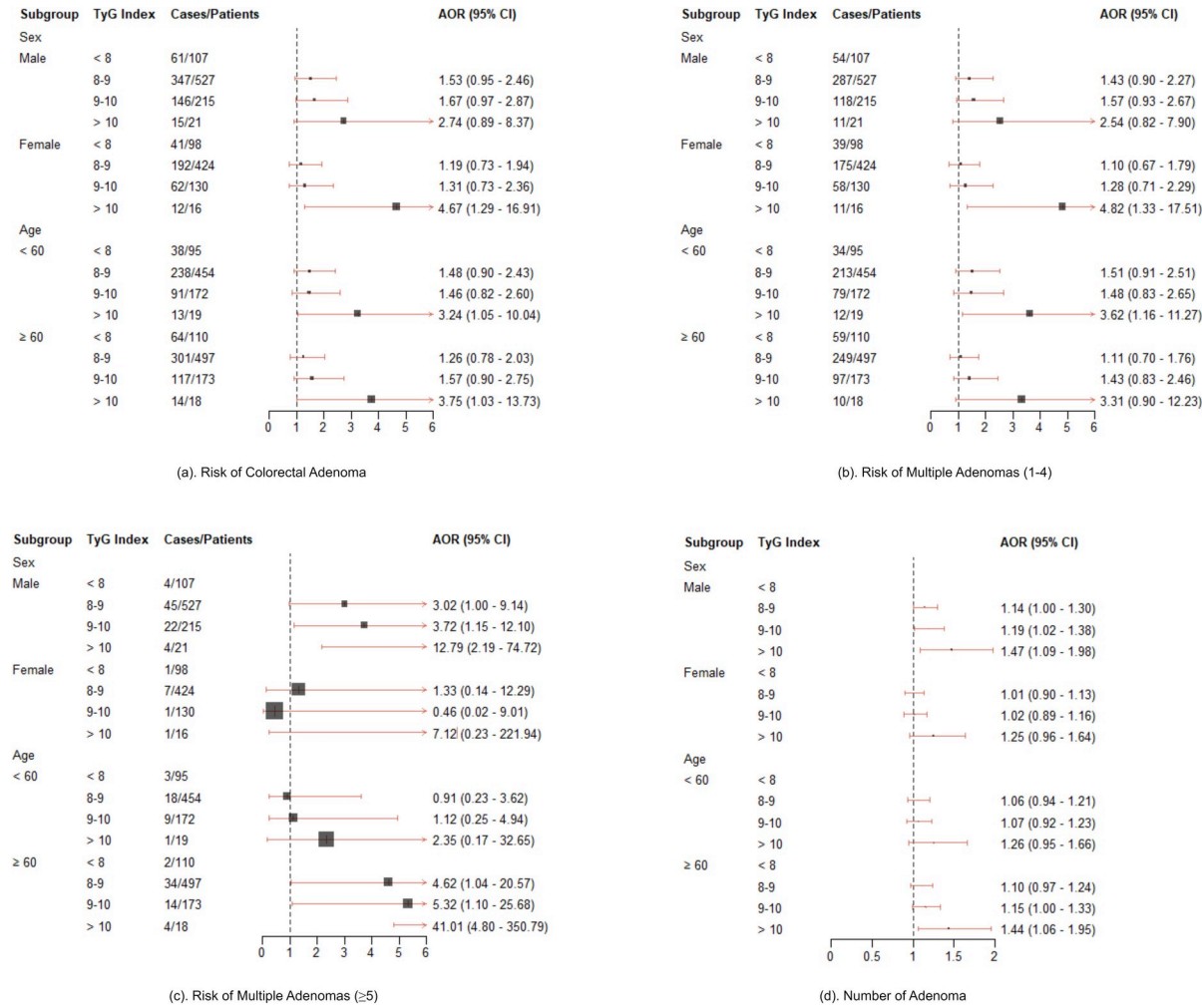

Fig 3. **Performance of TyG index across different subgroups.**

## Discussion

In this prospective cohort of 1,538 subjects who all underwent colonoscopies and blood tests for fasting blood glucose and triglycerides, we investigated the association of the TyG index on the incidence of colorectal adenoma and multiple adenomas ($\geq$5). We found that an increased TyG index is associated with a higher risk of incidence of adenoma and also multiple adenomas. We further explored that the effects grow non-linearly, as there was no significant effect for the middle TyG index group (TyG index 8–9); while the effect was the highest for the highest TyG index group. These non-linearly effects were consistently maintained in all analyses. Subgroup analyses showed that the association between the TyG index and risk of colorectal adenomas varied across gender, but there was no variability across age categories (<60 versus $\geq$60) in our study cohort, which might due to that this was an elder population of all subjects aged over 50 who underwent CRC screening. Moreover, the magnitudes of the estimates in multiple adenomas ($\geq$5) were substantially larger than merely detecting 1–4 adenoma(s). These findings suggest that the TyG index played a potential role to be a predictor of adenoma incidence and the number of adenomas.

To our knowledge, this is the first study that demonstrates the association between the TyG index and colorectal adenoma in a prospective cohort. In contrast to the two existing literatures [21, 22] that analyzed the association between the TyG index and CRC through retrospective data analysis, our prospectively collected data not only provides information on CRC detection but also includes data on adenoma detection and the number of adenomas detected. This comprehensive dataset allows us to explore potential links between insulin resistance and colorectal adenomas. Additionally, our findings are consistent with previous studies on insulin resistance and adenoma risk [15, 16]. Clinical studies have revealed that insulin resistance was associated with an increased risk of adenomas in both women [15] and men [43]. Insulin resistance can lead to hyperinsulinemia, which can directly activate insulin-like growth factor-1 (IGF-1) receptor, or inhibit IGF-binding protein (IGFBP) to increase the bioavailability of IGF-1 to the IGF-1 receptor [44]. The levels of IGF-1, the ratio of IGF-1/IGFBP-3, and insulin have been found to be associated with adenomas and advanced adenomas [45]. There also exists an interaction effect that central obesity and high triglyceride levels increase the risk of advanced and multiple adenomas [46]. Homeostatic Model Assessment of Insulin Resistance (HOMA-IR), has been proven to be a risk factor for colorectal adenoma in Japanese males [47]. As another valid biomarker for insulin resistance [17], our results show evidence that the TyG index is a complementary tool for adenoma risk evaluation.

Utilization of the TyG index can be easily implemented to risk-stratify subjects for early colonoscopy in clinical practice because of the readily available FBG and FTG blood results. FBG and FTG are common blood tests in primary care settings and the TyG index is calculated using a simple formula. Family medicine physicians and primary care practitioners may use this non-invasive TyG index and combine that with the patient's demographics, medical and family history to advise and refer those with a higher TyG index to early colonoscopy because of their higher risk of adenoma detection and multiple adenomas. Inferring from our results, subjects with TyG index >10 should refer to have early colonoscopy because of their increased risk of colorectal adenoma detection, as well as increased risk of multiple adenomas. Although the current study was not designed to determine the cost-effectiveness of the TyG index, the advantage behind this approach is readily apparent as the FBG and FTG blood tests are commonly arranged in primary care setting and their cost is generally low.

The major strengths of this study include the prospective study cohort of screening of MetS and CRC simultaneously, allowing the feasibility to integrate TyG index and CRC screening results in our study. Another advantage is the availability of prospectively collected potentially

confounding factors for adenoma, including cigarette smoking, alcohol drinking, overweight, existing diabetes, aspirin consumption, and FIT results, which are not always available in retrospective studies.

There are limitations in our study. Firstly, this is a single-centre study with solely Chinese subjects recruited. Future studies are necessary to confirm the generalizability of our study results across different populations. Secondly, this study consists of a CRC screening cohort from two different studies. Although the inclusion and exclusion criteria are the same, potential heterogeneity can be introduced because of different colonoscopy indications (direct screening regardless of FIT result in the 2008–2017 cohort versus positive FIT result in the 2018–2023 cohort). Lastly, the underlying mechanisms mediating the TyG index and colorectal adenoma risk remain unknown. Further studies are warranted to validate these results and explore the mechanisms.

In conclusion, subjects with elevated levels of TyG index, particularly those with TyG index >10, presented a higher risk of colorectal adenoma incidence and multiple adenomas. The findings in this study are helpful for stratifying asymptomatic subjects at high risk of adenoma and multiple adenomas for early colonoscopy in primary care settings.

## Author Contributions

**Conceptualization:** Thomas Yuen Tung Lam, Raymond Shing Yan Tang, Sunny Hei Wong, Rashid Nok Shun Lui, Simon Siu Man Ng, Samuel Yeung Shan Wong, Joseph Jao Yiu Sung.

**Data curation:** Ziyue Zhu, Thomas Yuen Tung Lam.

**Formal analysis:** Ziyue Zhu, Thomas Yuen Tung Lam.

**Funding acquisition:** Joseph Jao Yiu Sung.

**Investigation:** Thomas Yuen Tung Lam, Raymond Shing Yan Tang, Sunny Hei Wong, Rashid Nok Shun Lui, Simon Siu Man Ng, Samuel Yeung Shan Wong, Joseph Jao Yiu Sung.

**Methodology:** Thomas Yuen Tung Lam, Raymond Shing Yan Tang, Sunny Hei Wong, Rashid Nok Shun Lui, Simon Siu Man Ng, Samuel Yeung Shan Wong, Joseph Jao Yiu Sung.

**Supervision:** Joseph Jao Yiu Sung.

**Writing – original draft:** Ziyue Zhu, Thomas Yuen Tung Lam.

**Writing – review & editing:** Thomas Yuen Tung Lam, Joseph Jao Yiu Sung.

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
