## [Decision Letter · Decision Letter 0]

19 Aug 2024

PONE-D-24-18397Triglyceride-glucose index (TyG index) is associated with a higher risk of colorectal adenoma and multiple adenomas in asymptomatic subjects.PLOS ONE

Dear Dr. Lam,

Thank you for submitting your manuscript to PLOS ONE. After careful consideration, we feel that it has merit but does not fully meet PLOS ONE’s publication criteria as it currently stands. Therefore, we invite you to submit a revised version of the manuscript that addresses the points raised during the review process.

 to:Thomas Yuen Tung Lam

The manuscript you submitted (PONE-D-24-18397) was undergone external expert peer review and the first decision for your manuscript is: Minor changes.

At this point, you must provide point-by-point responses to the comments raised by the Reviewers.

We look forward to receiving your revised manuscript.

Kind regards,

Fernando Guerrero-Romero, MD, PhD

Academic Editor

PLOS ONE

Journal Requirements:

"The studies are fully funded by the Hong Kong Jockey Club Charities Trust (Ref. no.: 2018/0043). URL: https://charities.hkjc.com/charities/english/charities-trust/index.aspx"

Additional Editor Comments:

The manuscript you submitted (PONE-D-24-18397) was undergone external peer review and the first decision for your manuscript is: Minor changes.

At this point, you must provide point-by-point responses to the comments raised by the Reviewers.

Reviewers' comments:

Reviewer's Responses to Questions

**Comments to the Author**

1. Is the manuscript technically sound, and do the data support the conclusions?

Reviewer #1: Yes

Reviewer #2: Yes

2. Has the statistical analysis been performed appropriately and rigorously? 

Reviewer #1: I Don't Know

Reviewer #2: Yes

3. Have the authors made all data underlying the findings in their manuscript fully available?

Reviewer #1: No

Reviewer #2: Yes

4. Is the manuscript presented in an intelligible fashion and written in standard English?

Reviewer #1: Yes

Reviewer #2: Yes

5. Review Comments to the Author

Reviewer #1: The authors examine whether TgY Index is associated with risk of colorectal adenomas. They do a really nice job describing the statistical analysis methods used and where each method was used. Additionally, they do a good job with Tables 1-3. However, there is concern about the results sections labeled "Number of Adenoma" and "Subgroup Analysis".

For number of adenoma, it is not clear how the analysis was done. The paragraph mentions that the number of adenomas is continuous in nature. This would imply possibly a linear regression model was created. However, there is no mention of such a technique being used in the statistical methods section. Additionally, Table 4 reports odds ratios, which are not the result of running a linear regression. Thus, it appears that number of adenomas was treated as something other than continuous. It should be explicitly stated how number of adenomas was treated (continuous, ordinal, binary) and exactly which type of model was created for this analysis.

For the Subgroup analysis section, it is mentioned that both age and sex are found to be significant in the logistic and multinomial models. However, these results are not present anywhere else in the paper. It might be a good idea to add this information to the results presented in Tables 2 & 3 (and 4 if appropriate). This way, the reader can better follow along with the additional AOR's and see for themselves how the adjusted variables impact the results. The way the results of this section are presented appear different than the results of tables 2 & 3 (figure vs table). One style should be proposed for consistency. Additionally, it is not clear what is meant by " P for Interaction" from Figure 3. an interaction effect is usually one between two variables, and that does not appear to be what is presented. Clarification (or re-labeling) should be done for the p-value.

Reviewer #2: The current manuscript represents a secondary analysis of two cohort studies that evaluated the availability of the TyG index to identify the presence of colorectal adenoma and multiple adenomas in asymptomatic subjects ≥50 years.

In this regard, I have the following comments:

For consistency, I suggest changing the study's hypothesis by the objective in the abstract.

Figure 1 and its description should be moved to the Results section instead of Materials and Methods section.

It would be useful to determine the sensitivity, specificity, and positive and negative predictive values of the TyG index for detecting both colorectal adenoma and multiple adenomas. Please add.

6. PLOS authors have the option to publish the peer review history of their article (what does this mean?). If published, this will include your full peer review and any attached files.

Reviewer #1: No

Reviewer #2: No

---

## [Author Response · Author response to Decision Letter 0]

25 Aug 2024

Response to Reviewers:

Reviewer #1: The authors examine whether TgY Index is associated with risk of colorectal adenomas. They do a really nice job describing the statistical analysis methods used and where each method was used. Additionally, they do a good job with Tables 1-3. However, there is concern about the results sections labeled "Number of Adenoma" and "Subgroup Analysis".

For number of adenoma, it is not clear how the analysis was done. The paragraph mentions that the number of adenomas is continuous in nature. This would imply possibly a linear regression model was created. However, there is no mention of such a technique being used in the statistical methods section. Additionally, Table 5 (originally Table 4) reports odds ratios, which are not the result of running a linear regression. Thus, it appears that number of adenomas was treated as something other than continuous. It should be explicitly stated how number of adenomas was treated (continuous, ordinal, binary) and exactly which type of model was created for this analysis.

Thank you for your comments. We have added the following statement in “statistical analysis”:

“For the estimations using number of adenomas as the endpoint, since we had a heavy tail distribution for the number of adenomas, we transformed the dependent variable onto the log scale and applied log-normal regression.”

And we revised the design in abstract: “Its association with multiple adenomas (≥5) and the continuous number of adenomas were assessed by multinomial regression and log-normal linear regression, respectively.”

All “log-normal estimation” or “log-normal regression” have been revised as “log-normal linear regression” for clarification.

Table 4 reports estimates instead of odds ratios, as we stated in the “statistical analysis”: “The estimates would be interpreted as a percentage change in the number of adenomas.”

For the Subgroup analysis section, it is mentioned that both age and sex are found to be significant in the logistic and multinomial models. However, these results are not present anywhere else in the paper. It might be a good idea to add this information to the results presented in Tables 2 & 3 (and 4 if appropriate). This way, the reader can better follow along with the additional AOR's and see for themselves how the adjusted variables impact the results. The way the results of this section are presented appear different than the results of tables 2 & 3 (figure vs table). One style should be proposed for consistency. Additionally, it is not clear what is meant by " P for Interaction" from Figure 3. an interaction effect is usually one between two variables, and that does not appear to be what is presented. Clarification (or re-labeling) should be done for the p-value.

Thank you for your comments. The subgroup analysis for other endpoints has been added. The results are presented in the new Figure 3.

“P for Interaction” was meant for the test for the age/gender interaction with TyG index. To avoid the confusion, this item has been removed from the Figure. The individual significance level can be seemed from the 95% confidence interval in Figure 3.

Reviewer #2: The current manuscript represents a secondary analysis of two cohort studies that evaluated the availability of the TyG index to identify the presence of colorectal adenoma and multiple adenomas in asymptomatic subjects ≥50 years.

In this regard, I have the following comments:

For consistency, I suggest changing the study's hypothesis by the objective in the abstract.

Thank you for your comments. The title “objective” has been replaced by “hypothesis” accordingly.

Figure 1 and its description should be moved to the Results section instead of Materials and Methods section.

Thank you for your comments. Figure 1 and its description have been moved to the “Results” section.

It would be useful to determine the sensitivity, specificity, and positive and negative predictive values of the TyG index for detecting both colorectal adenoma and multiple adenomas. Please add.

Thank you for your comments. Table 3 (Table 3. Sensitivity, specificity, PPV and NPV for detecting adenoma and multiple adenomas) and the corresponding discussion have been added to the manuscript.

For adenoma detection: “The sensitivity, specificity, PPV and NPV of adenoma detection were presented in Table 3. Note that this model was not built for prediction and only aimed for showing the correlation between TyG index and risk of colorectal adenoma, and thus the values for sensitivity, specificity, PPV, and NPV were not sufficient to show enough discrimination power.”

For multiple adenomas detection: “The sensitivity, specificity, PPV and NPV of multiple adenomas detection were presented in Table 3. Even though the confusion matrix showed insufficient discrimination power, the Multi-class AUC was 0.70, presenting an acceptable discrimination power.”

---

## [Editor Report · Decision Letter 1]

3 Sep 2024

Triglyceride-glucose index (TyG index) is associated with a higher risk of colorectal adenoma and multiple adenomas in asymptomatic subjects.

PONE-D-24-18397R1

Dear Dr. Thomas Yuen Tung Lam

We’re pleased to inform you that your manuscript has been judged scientifically suitable for publication and will be formally accepted for publication once it meets all outstanding technical requirements.

Kind regards,

Fernando Guerrero-Romero, MD, PhD

Academic Editor

PLOS ONE

Additional Editor Comments (optional):

No additional comments
---

## [Editor Report · Acceptance letter]

16 Sep 2024

PONE-D-24-18397R1 

PLOS ONE

Dear Dr. Lam, 

I'm pleased to inform you that your manuscript has been deemed suitable for publication in PLOS ONE. Congratulations! Your manuscript is now being handed over to our production team.

Kind regards, 

on behalf of

Dr Fernando Guerrero-Romero 

Academic Editor

PLOS ONE